# Mesenchymal Stem Cell-Derived Apoptotic Bodies: Biological Functions and Therapeutic Potential

**DOI:** 10.3390/cells11233879

**Published:** 2022-12-01

**Authors:** Huixue Tang, Huikun Luo, Zihan Zhang, Di Yang

**Affiliations:** Liaoning Provincial Key Laboratory of Oral Disease, Department of Endodontics, School and Hospital of Stomatology, China Medical University, Shenyang 110002, China

**Keywords:** mesenchymal stem cells, apoptosis, apoptotic bodies, extracellular vesicles

## Abstract

Mesenchymal stem cells (MSCs) are non-hematopoietic progenitor cells with self-renewal ability and multipotency of osteogenic, chondrogenic, and adipogenic differentiation. MSCs have appeared as a promising approach for tissue regeneration and immune therapies, which are attributable not only to their differentiation into the desired cells but also to their paracrine secretion. MSC-sourced secretome consists of soluble components including growth factors, chemokines, cytokines, and encapsulated extracellular vesicles (EVs). Apoptotic bodies (ABs) are large EVs (diameter 500𠀓2000 nm) harboring a variety of cellular components including microRNA, mRNA, DNA, protein, and lipids related to the characteristics of the originating cell, which are generated during apoptosis. The released ABs as well as the genetic information they carry are engulfed by target cells such as macrophages, dendritic cells, epithelial cells, and fibroblasts, and subsequently internalized and degraded in the lysosomes, suggesting their ability to facilitate intercellular communication. In this review, we discuss the current understanding of the biological functions and therapeutic potential of MSC-derived ABs, including immunomodulation, tissue regeneration, regulation of inflammatory response, and drug delivery system.

## 1. Mesenchymal Stem Cells and Extracellular Vesicles

Mesenchymal stem cells (MSCs) are non-hematopoietic adult stem cells with self-renewal ability and multidirectional differentiation potential which express stem cell markers CD90, CD105, and CD73 [1]. MSCs were found initially in the bone marrow. So far, MSCs have been successfully isolated from a variety of tissues including dental pulp [2], periodontal ligament [3], adipose [4], synovial fluid [5], umbilical cord [6], and amniotic fluid [7]. MSCs have the ability to differentiate into osteoblasts [8], chondrocytes [9], adipocytes [10], vascular endothelial cells [11], cardiomyocytes [12], and neuronal cells [13]. Due to multidirectional differentiation potential, MSCs-based therapy seems superior in disease treatment. Additionally, MSCs also respond rapidly to cellular injury and migrate to the injury site to promote tissue repair and regeneration [14]. Accumulating evidence suggests that the therapeutic effects of MSCs are mainly attributed to their paracrine effects. Responding to stimulation factors, MSCs secrete growth factors, chemokines, cytokines, and other active factors to exert their pleiotropic actions [15], such as alleviating inflammation [16], regulating immune response [17], and promoting tissue repair [18].

Extracellular vesicles (EVs) are a group of lipid-bound vesicles that are released by various cells. According to the size and secretion mechanisms, EVs are divided into apoptotic bodies (ABs), microvesicles (MVs), and exosomes (Exos). ABs derived from MSCs (MSC-ABs) are of great distinction from exosomes. MSC-ABs (500–2000 nm) are produced by apoptotic MSCs which undergo plasma membrane deformation and sprout vesicle structures directly, but not membrane fusion. The enhanced hydrostatic pressure in apoptotic MSCs breaks organelles and other substances into fragments, which are packaged into different vesicles. Moreover, these membrane-bound vesicles are so-called MSC-ABs [19,20]. Exosomes (30–200 nm) evolving from the internal pathway of living cells were shown to be the naturally occurring nanospheres. Primely, extracellular substances fuse with intracellular endosomes through membrane invagination or endocytosis to form early endosomes. They start to mature and develop into late endosomes, followed by the formation of intraluminal vesicles, which fuse to form multivesicular bodies (MVBs). MVBs further fuse with the cell membrane and are released into the extracellular environment, termed exosomes (Exos) [19,20,21]. While MVs (100–1000 nm) originate from plasma membranes [22]. Due to the limitations of separation technologies, Exos are the most studied in EVs [23]. In contrast, the therapeutic application of ABs has been largely unexplored.

## 2. Apoptosis and Apoptotic Bodies

Apoptosis is widely known as programmed cell death. Unlike necrosis, apoptosis is an active and highly regulated process as a defense mechanism to maintain internal environmental homeostasis [24]. Apoptosis exhibits three distinct stages: (1) target cells receive death instructions and start programmed death; (2) a series of morphological and biochemical changes occur within the cell, such as cellular crumpling, chromatin disassembly, and nuclei condensation and (3) apoptotic cells or apoptotic bodies are removed by phagocytosis [25].

The initiation of apoptosis largely depends on caspases, which are highly critical effector enzymes in the apoptotic cascade reaction [25]. According to different effects, two types of caspases have been defined, the initiator caspase and execution caspase. Stimuli such as trauma, DNA damage, virus, and senility is a common cause of cell damage. Once cell damage is detected, the death signal can be transmitted through the signaling pathway to activate the initiator caspase and further activate the execution caspase, which gradually amplifies the apoptotic signal and eventually leads to apoptosis [26,27]. Two classic pathways exist in apoptosis: intrinsic and extrinsic. The intrinsic pathway is also called the mitochondrial apoptosis pathway. Mitochondria undergo osmotic translocation when damage factors act on cells, allowing apoptosis-inducing factor (AIF) between its inner and outer membranes to move into the cytoplasm, thus initiating apoptosis [28]. Unlike the intrinsic pathway, the extrinsic pathway is mediated by death receptors on the cell surface (DRs). Upon interaction with extracellular death ligands, DRs are activated and subsequently recruit adapter proteins such as TNF receptor-associated death domain (TRADD), Fas-associated death domain protein (FADD), and the apoptosis initiators cystathionin 8 and cystathionin 10. As signal conducting medium, these proteins bind with DRs to form the death-inducing signaling complex to induce apoptosis [25]. There are several common death receptors-ligands including Fas-FasL [29], TRAILR2-TRAIL [30], DR3-TL1A [31], and so on.

ABs are membrane structures released by cells during apoptosis. Breakage of apoptotic cells into ABs begins with the condensation of nuclear chromatin in cells, followed by cellular morphological changes which are mainly composed of membrane blebbing. Another characteristic change known as apoptotic volume decrease (AVD) often occurs at the early stage of blebs formation [32,33]. AVD refers to the loss of cell volume or cell shrinkage during apoptosis [34], which is the basis for forming cysts and the necessary geometric condition for constructing ABs. However, the mechanism and biological significance of AVD has not been entirely determined. AVD often occurs in two distinct stages. The first phase is the reversal of the normal sodium and potassium concentration gradients, resulting in intracellular sodium accumulation and potassium loss, which decreases cell volume to some extent [35]. Interestingly, during the secondary stage of AVD, there exists a premise that the cytoskeleton layout remains correct. Furthermore, this stage is accompanied by a continuous outflow of residual potassium, contributing to a further reduction in cell volume [36]. As an early event of apoptosis, AVD has an important effect on cell shrinkage and membrane vesicle formation [37]. Eventually, repeated blistering and contraction of apoptotic cells contribute to the formation of ABs packed with cellular contents [38]. It is noteworthy that different ABs produced by the same apoptotic cell may contain different contents, while a specific apoptotic body may contain multiple cellular components or even an intact organelle. The differences among components of ABs also determine the varied biological outcomes. In addition, there also exist some differences between MSC-ABs and Exos. For instance, Exos mainly contain proteins, lipids, mRNA and microRNA, but rarely DNA. However, MSC-ABs have a large content of DNA fragments, yet other components account for a relatively small proportion [39]. Although the formation of membrane vesicles appears a prerequisite for producing ABs, different cell types may exhibit different forms of membrane vesicles, such as microtubule spines, plasma membrane epithelium, and bead-like plasma membrane epithelium [40]. Apart from biological cargos distinction mentioned above, marker differences also exist between MSC-ABs and Exos. The Exos membrane contains a lipid raft structure composed of cholesterol, acantho myosin, and ceramide. Moreover, Exos also contain specific internal markers—tetraspanins, including CD63, CD81 and CD9, which are distinctive markers to identify Exos [41]. Exos are also rich in common proteins, including MVBs biogenetic proteins (Alix and TSG101), membrane transporters and fusion proteins (including RAB, GTPases), heat shock proteins (such as HSP60, HSP90). In addition, lipid-related proteins and phospholipase are frequently expressed in Exos [42]. It is worth noting that MSC-Exos not only express common surface markers like CD81 and CD9 but also carry MSCs markers CD44, CD73 and CD90 [43]. Unlike Exos described above, MSC-ABs are characterized by phosphatidylserine externalization, which contains several intracellular fragments and cellular organelles, including histones and fragmented DNA [44].

## 3. Biological Properties and Functions of Mesenchymal Stem Cell-Derived Apoptotic Bodies (MSC-ABs)

The disassembly of apoptotic cells into ABs has long been suggested to provide small fragments that can be easily engulfed by phagocytes to maintain the homeostasis of the internal environment. However, other ABs carry active factors and genetic materials to interact with target cells and alter the biological activities of downstream cells. Many studies highlighted the great potential for MSC-ABs in disease treatment, and these effects seem inextricably connected with their biological properties.

### 3.1. Immunomodulation

MSC-ABs inhibit the function of helper T 17 cells (Th17) or induce Th17 cells to transform into regulatory T cells (Treg), which leads to a decrease in the expressions of interleukin 22 (IL-22) and IL-23. Meanwhile, the anti-inflammatory factors Prostaglandin E2 (PGE2) and transforming growth factor-β (TGF-β) are increased. These cytokines regulated by MSC-ABs exert immunosuppressive or immunoenhancive effects on different immune cells, such as natural killer cells, dendritic cells, and T cells, etc. Thus MSC-ABs are verified to play critical roles in controlling the inflammatory immune response [45]. MSC-ABs have been shown to polarize macrophages to the targeted phenotype. Human bone marrow MSC-ABs induce macrophage polarization from M1 toward M2, thereby suppressing inflammation and promoting skin wound healing. In addition, MSC-ABs alleviated pulmonary bronchial dysplasia by promoting M2 polarization of macrophages [46]. Furthermore, human umbilical cord MSC-ABs (UCMSC-ABs) were found to activate M2-type macrophage polarization and thus promote wound healing in diabetic patients [47], as well as downregulate inflammatory symptoms in mice with colitis [48]. All of these indicate that MSC-ABs are of great importance in immune regulation.

Sepsis, a dysregulation of the immune response to infection [49], is often accompanied by infiltration of neutrophils and monocytes/macrophages. In severe cases, it may lead to multiple organ dysfunction, including liver and kidney [50]. Neutrophils exert a key influence on the immune system and inflammatory pathogenesis of sepsis. A special form of programmed cell death called NETosis exists in neutrophils [51] and generates neutrophil extracellular traps (NETs), which are composed of depolymerized chromatin and intracellular granule proteins [52], and play an important role in entrapping pathological stimuli, including bacteria fungi, protozoa, and virus. Previous studies have confirmed that treatment using apoptotic MSCs may abrogate the mortality of sepsis and may be superior to that of living cells [53], and these therapeutic effects are attributed to MSC-ABs. It was verified that MSC-ABs could be enriched in the bone marrow lumen of septic mice via electrostatic charge interactions with positively charged histones of NETs. More interestingly, MSC-ABs activated the Fas/FasL pathway [54] and switched the death pattern of neutrophils from netosis to apoptosis, thereby counteracting neutrophil infiltration and elevating survival in septic mice, even alleviating sepsis-induced organ dysfunction [55].

MSC-ABs are released by MSCs, and the “eat me” signal exposed on the surface keeps the same as that of apoptotic MSCs, which makes them easier to be recognized and swallowed by macrophages. Therefore, splitting into MSC-ABs contributes to a more-efficient clearance of apoptotic cells and seems important in controlling immune responses [25]. Compared with Exos, the unique clearance ability of MSC-ABs enables them to have better autoimmune regulation functions [56].

### 3.2. Promotion of Cell Proliferation and Tissue Regeneration

In a myocardial infarction (MI) rat model, MSC-ABs were corroborated to activate the lysosomes in endothelial cells (ECs) and increase the protein expression of lysosomal-associated membrane protein 1 (LAMP1) that maintains lysosomal structural integrity and function. At the same time, MSC-ABs mediated transcription factor EB (TFEB) translocation into the nucleus, which improved the angiogenic capacity of ECs and induced the autophagy of damaged cardiomyocytes. TFEB is a master molecule that regulates lysosomes and autophagy, which is tightly connected with angiogenesis and tissue regeneration [57]. In addition, the protein kinase B (PKB, also called AKT)-mediated vascular endothelial growth factor (VEGF) signaling pathway exhibits a higher level after treated by MSC-ABs, which promoted ECs regeneration and angiogenesis [58]. Taken together, MSC-ABs have the ability to improve cardiac function through increased cell autophagy and activation of the VEGF signaling pathway in a MI model. 

Protein translation elongation factors promote polypeptide chain elongation during mRNA translation [59]. In general, amino acids binding to amino-tRNA could be transported to the ribosome for protein synthesis [60]. Activation of mitochondrial translation elongation factor Tu (TUFM), one of the mitochondrial protein translation elongation factors, promotes amino acid delivery from the aminoacyl-tRNA to the mitochondrial ribosomal A site, thereby increasing the rate of amino acid elongation [61]. In vitro experiments substantiated that the transportation of TUFM from dental pulp stem cell-derived ABs (DPSC-ABs) into ECs activated the TFEB-mediated ECs autophagy via TFEB nuclear translocation in lysosomes. Then the translocated TFEB induced ECs autophagy and increased VEGF, angiotensin 2 (ANG2), matrix metalloproteinase2 (MMP2), and other angiogenesis-related factors, which are strongly linked to the angiogenic abilities of ECs. In contrast, the knockdown of TUFM in DPSCs resulted in a corresponding decrease in both the autophagy and angiogenesis ability of ECs. The same mechanism was also found in the animal model. In in vivo experiments, DPSC-ABs specifically activated endogenous EC autophagy, and further inducing EC biological behavior and causing angiogenesis. Ultimately, the accelerated revascularization stimulated the regeneration of the dental pulp. Furthermore, compared with the control group, the DPSC-ABs-treated group showed thickened dentin and elongated root [62].

Several studies have confirmed that BMSCs could be applied to repair and regenerate bone defects. However, most BMSCs experienced considerable death after their transplantation, causing decreased therapeutic effects. The emergence of ABs has compensated for this deficiency to some extent. Indeed, after transplantation into defects of the cranium 2 days, BMSCs undergo apoptosis and release ABs. Uptake of BMSC-ABs apparently improves the proliferation, migration, and differentiation of endogenous BMSCs, consequently enhancing the homing effect. Meanwhile, BMSC-ABs act as c-Jun N-terminal kinase (JNK) signaling agonists by increasing intracellular ROS levels, ultimately leading to new bone formation and bone tissue regeneration [63].

Beyond its function in bone regeneration, MSC-ABs also promote skin and mucosal wound healing through different pathways and modalities. Upon the stimulation of tumor necrosis factor-α (TNF-α), MSC-ABs promoted gingival wound healing via the Fas/Fap-1/Cav-1 axis [64]. In addition, BMSC-ABs also perform a reparative effect on skin injury [65]. Under BMSC-ABs treatment, skin-injured mice showed faster healing and a more integrated skin structure with newly formed epithelium, hair follicles, and collagen at the trauma site. These reparative effects may be attributed to the indirect effect on fibroblasts. MSC-ABs have been exemplified to promote the polarization of macrophages to M2 type, which further enhances fibroblast proliferation and migration, and further facilitates skin wound healing. In addition, MSC-ABs increased the metabolic level in mice by activating the Wnt/β-catenin pathway in skin and hair follicles MSCs, consequently achieving wound healing and hair growth [66].

### 3.3. Regulation of Inflammatory Response

Recently, apoptotic bodies from human adipose tissue mesenchymal stem cells (ADMSC-ABs) were reported to reduce pathological symptoms in a mouse model of atopic dermatitis, which revealed the ability of MSC-ABs in inflammation regulation. ADMSC-ABs produced remarkable effects not only on reducing the release of inflammatory cytokines such as interferon γ (IFN-γ), thymic stromal lymphopoietin (TSLP), and IL-4 but also on decreasing the infiltration of inflammatory dendritic epidermal cells. Thus, the pathological symptoms were relieved and the skin wound also got healed [67,68]. In addition, in vivo studies demonstrated that MSC-ABs reduced the level of inflammatory cytokines IL-1β, IL-18, and TNF-α by downregulating the expression of NF-kB p65, C-C motif chemokine 17 (CCL-17) and CCL-24, eventually alleviating trinitrobenzene-sulfonic acid (TNBS)-induced experimental colitis [69,70].

Macrophages play an essential role in the inflammation progression of periodontal disease. Whereas phagocytosis of BMSC-ABs by macrophages inhibits the secretion of tumor necrosis factor-α (TNF-α) and IL-6 by reducing the expression of cyclooxygenase-2 (COX2) in inflammatory macrophages, which in turn inhibits the expression of tartrate-resistant acid phosphatase (TRAP) and matrix metalloproteinase 9 (MMP-9). Meanwhile, the reduction in COX2 by BMSC-ABs increased the level of IL-10 in a dose-dependent manner, which further suppressed osteoclast differentiation and bone resorption. In addition, BMSC-ABs inhibited the polarization of macrophages skew into pro-inflammatory phenotypes via AMPK/SIRT1/NFkB pathway [71]. To summarize, all these results suggest that MSC-ABs are of great significance in alleviating porphyromonas gingivalis lipopolysaccharide (Pg.-LPS)-induced macrophage inflammation as well as periodontitis.

### 3.4. Drug Delivery System

Compared with other extracellular vesicles, ABs-based drug carrying and delivery remains poorly understood. However, ABs have been shown an emerging potential as drug carriers recently. In terms of their structure characteristics, ABs have a lipid bilayer membrane structure that seems similar to cell membranes and has low toxicity. In addition, ABs transfer different molecules such as DNA, RNA and proteins, thereby effectively regulating various aspects of the recipient cell [72]. Fibroblast-derived ABs transferred the proto-oncogene c-myc to recipient cells, resulting in a tumorigenic phenotype in vivo [73].

As a complex and strict biological barrier, the blood–brain barrier can keep most biological factors and drugs from the brain, which maintains the stability of the internal environment of brain tissue. It is found that anti-TNF-α antisense oligonucleotide (ASO) combined with cationic konjac glucomannan (cKGM) to form cKGM/ASO complex (CKA), which seems easier to transfect into cancer cells. This complex can be easily loaded into B16F10 cells (brain metastatic cancer cells)-derived small apoptotic bodies (sABs) via transfection. CKA-loaded sABs which seem abundant of ASO were transcytosed by bEnd3.cell (brain microvascular endothelial cell, BMEC) to penetrate the blood–brain barrier and finally be absorbed by microglial cells in the brain. In the Parkinson’s (PD) mouse model, CKA-loaded sABs significantly improved PD symptoms through the anti-inflammatory effect of ASO. These results indicate the possibility of brain-targeted drug delivery by ABs derived from brain metastatic cancer cells and reveal that ABs have gradually become excellent drug carriers because of their high efficiency of delivery and large-scale production potential [74]. Chimeric apoptotic bodies (cABs) are delivery systems consisting of natural membranes of T-cell ABs and mesoporous silica nanoparticles (MSN). cABs achieve targeted regulation of inflammation by pre-loading anti-inflammatory drugs on MSNs, which were delivered and released to the inflammation site through the natural membrane of ABs. It has been validated that cABs can target and modulate macrophages toward the M2 phenotype to attenuate skin inflammation and enteritis [75]. These results indicate that MSC-ABs, one component of ABs, may also have drug delivery potential.

MSC-ABs transferred the ubiquitin ligase RNF146 (RING finger protein 146) and miR-328-3p to target Axin1 in MSCs and rescued the impaired stem cells by activating the Wnt/β-catenin pathway [76]. More importantly, the properties of MSC-ABs can be adjusted by pretreatment of the MSC culture process in order to reach specific therapeutic effects, such as adding chemokines or cytokines, pretreating with small molecule drugs, introducing gene modifications, etc. [77].

However, some other concerns should nevertheless be raised regarding the direct use of ABs as a drug carrier, such as immune rejection, and midway degradation, the latter can be remedied by biomaterials. MSC-ABs encapsulated in hyaluronic acid (HA) hydrogel for intrauterine infusion were reported to be effective in relieving intrauterine adhesions [78]. By integrating MSC-ABs with a HA hydrogel, the MSC-ABs delivery system was established, in which HA hydrogel promoted MSC-ABs retention and facilitated their continuous release. In a murine model of acute endometrial damage and a rat model of IUAs, transplantation of MSC-ABs-laden HA hydrogel into uteri induced M2-type macrophage polarization and promoted endometrial cell proliferation and blood vessel formation, thereby significantly increasing endometrial thickness and the number of endometrial glands, reducing the fibrosis, and ultimately enabling the recovery of fertility. In this process, HA hydrogel only promoted the retention of MSC-ABs at the site of interest and acted as a separator to prevent uterine adhesions, all of which enhanced the therapeutic effect of MSC-ABs.

## 4. Summary

Recently, MSCs have attracted considerable attention because of their easy accessibility, high differentiation ability, and fast renewal rate. MSC-ABs, as their products, can avoid the increase in apoptosis generated by cell transplantation and some ethical issues. Besides, the application of MSC-ABs in drug delivery, and vaccine development could also as a promising candidate for developing cell-free therapy.

However, there are still many aspects that should be outlined concerning the use of MSC-ABs. Firstly, MSC-ABs may have different biological properties from MSCs and may play heterogeneous roles in disease treatment. Secondly, the target specificity and toxicity strength of MSC-ABs still cannot be fully determined, and the safety also needs to be further explored. Thirdly, ABs isolated from different MSCs may have different regulatory effects. Therefore, the synergy or antagonism arising from the interaction between different ABs needs to be well understood in practical applications [79,80]. Taken together, the safety and stability of MSC-ABs, pretreatment strategies, drug delivery methods, as well as targeting effects will become new directions for future research and provide more guiding suggestions for clinical applications.

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
