# Peer review of "Mesenchymal Stem Cell-Derived Apoptotic Bodies: Biological Functions and Therapeutic Potential"

_cells, 2022, doi:10.3390/cells11233879_

Round 1

Reviewer 1 Report

Please provide evidence to compare exosome and MSC-derived apoptotic bodies in terms of their biomarkers, functionality and related mechanism

Author Response

Dear reviewer,

Thank you for your comments and suggestions about the article“Mesenchymal stem cell-derived apoptotic bodies: biological functions and therapeutic potential”. Your suggestions have enabled us to improve our work. And the responses are in the attachment.

Reviewer 2 Report

In the manuscript by Tang et al, “Mesenchymal stem cell-derived apoptotic bodies: biological functions and therapeutic potential”. In this manuscript, authors have described the biological functions and therapeutic potential of MSC-apoptotic bodies, one of the types of extracellular vesicles.

Based on the literature, these apoptotic bodies are considered as debris and are easily removed during isolation of exosomes (most studied EVs sub types).

As these are least studied in the EVs, authors must collect more detailed information for them to considered as therapeutic potential: such as how they differ from exosomes with respect to biological cargos?

A very similar manuscript has been recently published on Apoptotic bodies of MSCs, which is very close to this manuscript. Link below:

https://doi.org/10.3390/cimb44110351

So, I don’t feel that the current manuscript in the current format can be published.

Authors need to work out for this with respect to biological cargos responsible for therapeutic potential of MSCs apoptotic bodies. 

Author Response

(The authors gave the same response as above.)

Round 2

Reviewer 2 Report

Authors have addressed the majority of the comments, so it is recommended to accept.